# Effect of Al_2_O_3_ Sandblasting Particle Size on the Surface Topography and Residual Compressive Stresses of Three Different Dental Zirconia Grades

**DOI:** 10.3390/ma14030610

**Published:** 2021-01-28

**Authors:** Hee-Kyung Kim, Byungmin Ahn

**Affiliations:** 1Department of Prosthodontics, Institute of Oral Health Science, Ajou University School of Medicine, Suwon 16499, Korea; 2Department of Materials Science and Engineering and Department of Energy Systems Research, Ajou University, Suwon 16499, Korea; byungmin@ajou.ac.kr

**Keywords:** zirconium oxide, air abrasion, dental stress analysis, phase transition, surface properties

## Abstract

This study investigated the effect of sandblasting particle size on the surface topography and compressive stresses of conventional zirconia (3 mol% yttria-stabilized tetragonal zirconia polycrystal; 3Y-TZP) and two highly translucent zirconia (4 or 5 mol% partially stabilized zirconia; 4Y-PSZ or 5Y-PSZ). Plate-shaped zirconia specimens (14.0 × 14.0 × 1.0 mm^3^, n = 60 for each grade) were sandblasted using different Al_2_O_3_ sizes (25, 50, 90, 110, and 125 μm) under 0.2 MPa for 10 s/cm^2^ at a 10 mm distance and a 90° angle. The surface topography was characterized using a 3-D confocal laser microscopy and inspected with a scanning electron microscope. To assess residual stresses, the tetragonal peak shift at 147 cm^−1^ was traced using micro-Raman spectroscopy. Al_2_O_3_ sandblasting altered surface topographies (*p* < 0.05), although highly translucent zirconia showed more pronounced changes compared to conventional zirconia. 5Y-PSZ abraded with 110 μm sand showed the highest Sa value (0.76 ± 0.12 μm). Larger particle induced more compressive stresses for 3Y-TZP (*p* < 0.05), while only 25 μm sand induced residual stresses for 5Y-PSZ. Al_2_O_3_ sandblasting with 110 μm sand for 3Y-TZP, 90 μm sand for 4Y-PSZ, and 25 μm sand for 5Y-PSZ were considered as the recommended blasting conditions.

## 1. Introduction

Structural ceramics have been applied in dentistry to replace the metallic dental prostheses due to the increasing interest in esthetics and growing concerns about metal allergies [1]. Among the dental ceramics, zirconia ceramics are rapidly gaining popularity which may have excellent mechanical properties, high biocompatibility, and relative esthetic potential [2]. Three phases of zirconia are exhibited depending on the temperature: monoclinic below 1170 °C, tetragonal between 1170 and 2370 °C, and cubic above 2370 °C [3]. Since a volumetric change of about 3–5% is accompanied during the phase transformation from the tetragonal to monoclinic phases, a dopant oxide such as yttria (Y_2_O_3_) is used to stabilize a tetragonal structure at room temperature [3]. The 3 mol% yttria-stabilized tetragonal zirconia polycrystal (3Y-TZP) with high mechanical strength has been widely used for dental applications and can be applied for the fabrication of crowns, veneers, fixed partial prostheses, dental implant, and orthodontic brackets combined with computer-aided design/computer-aided manufacturing (CAD/CAM) technologies [4]. 

Although zirconia has been considered a promising biomaterial in dentistry, it has some drawbacks for the clinical application of this material. For example, 3Y-TZP has certain optical disadvantages due to its relatively high refractive index, which causes a high grade of light reflection [5]. The high reflectability leads to a further deterioration in translucency and therefore esthetics. Hence, recent developments have focused on improving the translucency of 3Y-TZP, especially as a monolithic (full anatomical) system for dental restorations. More recently, novel highly translucent zirconia has been introduced by the addition of an increased amount of stabilizing oxide (Y_2_O_3_) to stabilize a cubic structure at room temperature [6]. Although their reduced mechanical properties are generally thought to be a major weak point, the enhanced translucency of cubic-phase containing partially stabilized zirconia ceramics (4 mol% partially stabilized zirconia; 4Y-PSZ and 5 mol% partially stabilized zirconia; 5Y-PSZ) could allow the fabrication of esthetic restorations without the need for further veneering techniques [6,7]. 

Another potential concern with zirconia restorations is the problematic bonding of resin cements to zirconia, which is associated with the clinical reliability. Zirconia is a non-silica-based ceramic and thus, zirconia surfaces cannot be etched with hydrofluoric acid [8]. The retention of zirconia restorations depends on the mechanical roughening of the surface and the chemical bonding with an acidic adhesive monomer such as 10-methacryloxydecyl dihydrogen phosphate (MDP) [9]. The most commonly used mechanical pretreatment to promote the bonding of resin cement to zirconia is the sandblasting of the interior surface of zirconia restorations with aluminum oxide (Al_2_O_3_) particles [10]. The sandblasting technique can provide a clean and rough surface for micromechanical interlocking [11]. In addition, the sandblasting can be effective to increase the surface area, surface wettability, and therefore the surface energy [11]. However, the bonding performance would be highly dependent on the sandblasting conditions, such as the size and shape of abrasive particles [12,13], air pressure [12,14], working duration [15], distance [14,16], and impact angle [16]. The sandblasting tends to trigger the tetragonal to monoclinic phase transformation of conventional high-strength zirconia (3Y-TZP) [17], whereby the transformation toughening mechanism [2] can lead to increased mechanical strengths [14]. However, excessive sandblasting may cause damage to the zirconia surface, deteriorating the flexural strength of 3Y-TZP [12,13,14,18]. 

Surface topography can influence the geometrical, physical, and chemical properties of the material [19]. For the measurement of the surface topography, quality control as well as the prediction of the surface functional properties can be crucial factors. In the analysis of surface texture, areal based three-dimensional analyses, defined within ISO 25178 [20], can provide more precise information than two-dimensional profile measurements. A number of studies investigated the characterization methods for surface topographies. The feature-based characterization, a recently introduced method, could offer spatial information in relation to the orientation of topographical properties of the surfaces [20]. Another systematic approach is multiscale analyses, which include the characterization of surface topographies at multiple scales of observation to control the different interactions of the scales [19]. A new 3-D-motif method can be applied to determine the geometric features like holes and valleys which are created by sandblasting [21]. Reflectance confocal microscopy based on the focus detection method is one of the optical metrology techniques. Noncontact optical measurement methods have been widely used for their great reliability and flexibility, but the noise of the light source can affect the measurement quality [22]. A novel spiral-scanning laser differential confocal measurement method can help reduce the existing disturbance [23].

It is particularly significant to assess the geometry of abrasive grains and the determination of their machining potential which could be related to the interaction of abrasive grains with material’s surface texture. Thus, the accurate measurements of the topography of abrasive grains affect the efficiency of the machining process. Many studies suggested sets of parameters for the quantitative assessment of the surface geometry. Qiao et al. applied sets of 3-D parameters (amplitude, spatial, hybrid, and functional parameters according to ISO 25 178) to characterize the active surface of abrasive tools due to the lack of accuracy with a single parameter [24]. Recent studies dealt with new parameters in order to increase the machining potential. Kacalak et al. used two new parameters (Shs and Shos) to determine the state of the active surface of abrasive tools considering the shape of abrasive grains, the orientation of their cutting edges, and the sharpness of protrusions during the abrasive process [25]. Recently, Gogolin et al. described the efficiency of fluid flow parameters to decrease heat transfer during the erosive process [26]. Pawlus et al. discussed the Rk family parameters which was based on the material ratio curve [27]. They reported that Rk family parameters could be easily calculated but led to errors in the interpretation. 

The volume difference during the t→m phase transformation could create compressive residual stresses in the zirconia surface [17] and those residual stresses would be beneficial to improve the flexural strength of 3Y-TZP [28]. Confocal Raman spectroscopy using laser microprobes has been widely used to quantify the residual stresses in the zirconia surface non-destructively [29,30]. The induced residual stresses trigger the phonon deformation of the Raman bands and thus, the wavenumbers of zirconia crystals were altered [31]. The amount of band shift could be used to measure residual stresses related to the t→m phase transformation. Several studies [29,32,33] demonstrated that the tetragonal peak at around 147 cm^−1^ shifted towards a higher wavenumber, indicating the presence of residual compressive stresses. A recent study measured the stress-induced Raman peak shift of the metastable tetragonal phase of yttria-stabilized zirconia at 147, 465, 610, and 640 cm^−1^ when annealing at high temperature [30]. The authors attributed the Raman peak shift to the structural changes of zirconia crystals. They proposed that the internal strain mismatch caused compressive stresses and as a result, enhanced the material’s toughness. As highly translucent dental zirconia and conventional zirconia have different phase compositions and microstructures [7], they may behave differently to sandblasting. Several studies compared the sandblasting effects between different zirconia grades. Zhao et al. reported that the changes in surface roughness of highly translucent zirconia were obtained faster than those of conventional zirconia under the same sandblasting conditions [10]. Sandblasting treatments contributed to the durable bond of resin cement to highly translucent zirconia [34], although sandblasted translucent zirconia exhibited a lower bond strength compared to the sandblasted conventional zirconia [35]. On the contrary, the mechanical sandblasting did not significantly increase the surface roughness of highly translucent zirconia, while the abrasive blasting changed its phase compositions in the study by Inokoshi et al. [33]. Inokoshi et al. also detected a change in the Raman peak of the tetragonal zirconia band at 146 cm^−1^ in the sandblasted highly translucent zirconia. However, these studies were conducted under different sandblasting parameters. A recent study reported that the flexural strength of sandblasted highly translucent zirconia decreased with increasing air-abrasion pressure [36]. In that study, only 50 μm alumina sand was employed and the flexural strength was measured using a three-point bending test with the loading on the blasted surface. 

Appropriate sandblasting protocols on the durable bond between resin cement and highly translucent zirconia have not well been studied yet. Moreover, most of the previous studies on the various effects of alumina sandblasting have used only a few particle sizes; 50 μm [14,18,33,34,35,36], 50 and 110 μm [9,12,15], 110 μm [16], 150 μm [13], 110 and 250 μm [17]. This study focused on the influence of the sandblasting particles with five different sizes on the sandblasting performance of dental monolithic zirconia with three different levels of translucency, while other sandblasting parameters were kept constant. We used commercially available alumina particle sizes on the market and tried to determine any changes in the sandblasting effects as a function of particle size. In this study, we investigated the 3-D surface topographies by using a confocal laser scanning microscopy, which is a noncontact optical imaging technique based on capturing multiple 2-D images at different depths which enables the reconstruction of a 3-D structure through an optical sectioning process. In addition, the Raman peak shift at 147 cm^−1^ was used to feature the residual compressive stresses under sandblasting with different alumina particle sizes, which can be a significant parameter related to the mechanical properties of zirconia. The purpose of this study was, therefore, to elucidate the effect of Al_2_O_3_ sandblasting particle size on the surface topography and residual compressive stresses of conventional tetragonal zirconia (3Y-TZP) and two different grades of highly translucent zirconia (4Y-PSZ and 5Y-PSZ). More importantly, this study tried to offer clinical guidelines in selecting the optimal sandblasting particle to achieve the required surface topography with minimal adverse effects in novel dental zirconia materials. The null hypotheses tested in this study were that the sandblasting with a different Al_2_O_3_ particle size would neither modify surface topographies nor induce any residual stresses in the surface zone of monolithic dental zirconia, and that those responses to sandblasting would not significantly differ among the three zirconia grades investigated. 

## 2. Materials and Methods

The experimental study design was schematically depicted in Figure 1; specimen preparation and Al_2_O_3_ sandblasting conditions as well as the analytical methods associated with surface topography, residual stresses, microstructure, and Al_2_O_3_ particle analysis are presented.

### 2.1. Specimen Preparation

The dental zirconia materials with three different levels of translucency investigated in this study are described in Table 1. 

A total of 180 zirconia specimens were prepared in the form of sintered plates (14.0 mm × 14.0 mm × 1.0 mm, n = 60 for each grade). One side of the specimen was sequentially polished through 800-grit silicon carbide papers under running water to ensure identical initial roughness, and then all specimens were thermally etched at 1400 °C for 30 min. The specimens of each zirconia grade were randomly divided into six subgroups of 10 specimens. One subgroup of each zirconia grade was kept as-polished (untreated, control) and the others were sandblasted.

### 2.2. Sandblasting

Sandblasting was carried out using commercially available Al_2_O_3_ particles of five different sizes (25, 50, 90, 110, and 125 μm; Cobra, Renfert, Hilzingen, Germany), while other parameters were kept constant. The zirconia plates were mounted in a custom-made specimen holder designed by the authors (Figure 1) at a distance of 10 mm from the tip of the sandblasting unit equipped with a 2.0 mm diameter nozzle. Specimens were then particle-abraded at 0.2 MPa pressure, for 10 s/cm^2^, at 90° impact angle using a sandblasting device (Basic master, Renfert, Hilzingen, Germany). Only the polished side received the sandblasting treatment. After sandblasting, all specimens were ultrasonically cleaned in 99.8% isopropanol for 20 min at a frequency of 40 kHz to remove the debris of Al_2_O_3_ particles from the zirconia surface, and then air dried.

### 2.3. Abrasive Blasting Particle (Al_2_O_3_) Analysis

Since the quality of the particle size distribution determined the significance and reliability of the analysis, the granulometric distribution of five different sizes of Al_2_O_3_ blasting particles were measured through a wet granulation process with a laser scattering particle size analyzer (LSPSA; LA-350, HORIBA, Kyoto, Japan). Distilled water was used as a binding liquid. The specific surface area of the investigated particles was in intervals from 0.1 to 1000 μm and the measuring time was 10 s.

The trace elements in the Al_2_O_3_ particles were verified by inductively coupled plasma optical emission spectrometry (ICP OES; Agilent 5100, Agilent, Santa Clara, CA, USA) with a charge injection device (CID) detector. The specimens were introduced with the OneNeb Series 2 inert concentric nebulizer (Agilent, Santa Clara, CA, USA) and an inert double-pass spray chamber with ball joint socket. The TOPEX microwave digestion system (PreeKem, Shanghai, China) was used for centrifugation. 

The morphological images of the Al_2_O_3_ abrasive particles were acquired by using a scanning electron microscopy (SEM; JSM-IT500HR, JEOL, Tokyo, Japan) at magnifications of 50×, 100×, and 200×. The acceleration voltage of the cathode was set to 15 kV. All particles were platinum (Pt) coated before SEM examinations.

### 2.4. Surface Topography Characterization

Surface topography was investigated on the specimens of each subgroup using a three-dimensional (3-D) confocal laser scanning microscopy (CLSM; LEXT OLS3000, Olympus, Tokyo, Japan) at 50× magnification. The areal texture parameters were obtained using a software (LEXT-OLS, version 6.0.3, Olympus, Tokyo, Japan): Sa, the arithmetic mean height; Sq, the root mean square height; and Sv, the maximum pit height of the scale-limited surface according to ISO 25178 [20]. Surface measurements were processed with the form and outlier eliminated. Tilt was corrected and a 3-D surface was constructed with the distance to the optical center in the X axis, the tilt angle on the Y axis, and the flatness error on the Z axis. A robust short wavelength pass Gaussian filter (cut-off wavelength: 10 μm) was applied to the data in order to decompose waviness from roughness. For each specimen, three different measurements (effective field of view was 256 × 192 μm) on the either polished sides for controls or sandblasted sides for experimental subgroups were performed. A total of 30 measurements was obtained for each subgroup. 

### 2.5. Microscopic Surface Structure and Elemental Composition 

The surface microstructures were observed under a scanning electron microscope (JSM-7800F Prime, JEOL, Tokyo, Japan) equipped with an energy-dispersive X-ray spectroscopy (EDX; Inca, Oxford Instruments, Abingdon, UK). One specimen from each subgroup was randomly subjected to SEM at 3000× and 10,000× magnifications. The acceleration voltage of the cathode was set to 5.0 kV and the working distance (WD) was 10.0 mm. After the specimens were sputter-coated with gold, secondary electron SEM images were acquired in vacuum (10^−5^ mbar). The elemental compositions of the specimen surface were analyzed using EDS.

### 2.6. Phase Transformation and Compressive Residual Stress

Micro-Raman spectroscopy (μRaman) was used to identify the phase transformation and surface residual stresses induced by sandblasting. Raman spectra (LabRAM HR Evolution, Horiba Scientific, Kyoto, Japan) were collected using a 532 nm wavelength diode-pumped solid-state laser (DPSSL) of 10 mV through a 100× objective with a pinhole aperture of 50 μm. The collection time of Raman scattering was 10 s and two consecutive spectra were averaged. For each subgroup, 25 measurements were obtained. For a quantitative determination of residual stresses within the specimens, the Raman wavenumber of the tetragonal zirconia (t-ZrO_2_) band at around 147 cm^−1^ was traced [29,32,33] using a curve fitting software (Origin 2020 Pro, OriginLab Corp., Northampton, MA, USA).

### 2.7. Statistical Analysis

All tests were performed at a significance level of *α* = 0.05 using a statistical software (IBM SPSS Statistics for Windows, v25.0, IBM Corp., Chicago, IL, USA). A normal distribution and homogeneity of variances were verified with Shapiro–Wilk test and Levene test, respectively (*p* < 0.05). The statistical significant differences among the various blasting particle sizes were analyzed with a one-way analysis of variance (ANOVA) followed by Tukey’s honestly significant difference (HSD) post hoc test. A two-way ANOVA was performed to determine the effect of two independent variables (yttria content; 3 mol% yttria for 3Y-TZP, 4 mol% yttria for 4Y-PSZ, and 5 mol% yttria for 5Y-PSZ and an alumina particle size) (1) on the surface topography and (2) on the Raman wavenumber of the tetragonal zirconia (t-ZrO_2_) band at around 147 cm^−1^. The interactions between two independent variables were verified and pairwise comparisons for simple main effects of independent variables were analyzed by using SPSS syntax. In addition, the Pearson correlations between the particle size and surface texture parameters of the subgroups for all zirconia grades were analyzed. 

## 3. Results

### 3.1. Al_2_O_3_ Particle Size, Morphology, and Chemical Composition

Figure 2 demonstrated the particle size characteristics of five different specifications of the Al_2_O_3_ particles. There were significant differences among all specifications and the measured particle sizes increased concomitantly with the increased specification (*p* < 0.05). The specification of 125 μm had the widest particle size distribution ranging from 29.91 to 592.39 μm, while the narrowest particle size distribution, ranging from 13.25 to 116.2 μm, was for the 25 μm specification. 

Figure 3 showed the particle size distributions and microstructures of Al_2_O_3_ particles. The SEM images revealed that the finest particles were observed in the 25 μm specification, while the largest particles prevailed in the 125 μm specification. Most of the particles had grit shapes with sharp and irregular corners and only a few of them were spherical in shape. The particles contained structural flaws, such as cracks and micro-cracks, and a higher number of defects was observed in the larger particles. It was observed that the particles of 125 μm specification were loosely packed, while the particles of 25 μm specification were homogeneously distributed. 

Table 2 showed that the concentrations of trace elements in the Al_2_O_3_ particles obtained by ICP OES. Si, Fe, and Zr were detected at variable amounts. The 125 μm alumina sand had the highest amounts of these elements with 545.963 ± 3.71, 80.348 ± 0.78, and 5.645 ± 0.01 mg/kg for Si, Fe, and Zr, respectively. 

### 3.2. Surface Topographic Properties 

The representative CLSM images and SEM photomicrographs after sandblasting using different sizes of Al_2_O_3_ for three different dental zirconia grades are shown in Figure 4, and the values of Sa, Sq, and Sv parameters of each subgroup are shown in Figure 5. As shown in Figure 4 and Figure 5, the surface roughness significantly increased with an increase in the particle size up to 110 μm, whereas the surface roughness values decreased with 125 μm alumina sand, lying between those with 50- and those with 90 μm alumina sand for all zirconia grades (*p* < 0.05). There was a statistically significant interaction between the Y_2_O_3_ amount and Al_2_O_3_ particle size on the surface topography of the zirconia specimens based on the result of a two-way ANOVA (*p* < 0.05). Simple main effects analyses showed that the Sa and Sq parameters were not significantly changed with 25 μm alumina sand in most subgroups (*p* > 0.05), and that there were no significant differences in the scalar values (Sa) among the non-abraded subgroups (3Y-con, 4Y-con, and 5Y-con) and among the 25 μm abraded subgroups (3Y-25, 4Y-25, and 5Y-25) (*p* > 0.05). For the sandblasted 4Y- or 5Y-PSZ, higher scalar values (Sa, Sq, and Sv) were acquired compared to those of sandblasted 3Y-TZP with the same particle size. 5Y-PSZ abraded with 110 μm sand showed the highest Sa values (0.76 ± 0.12 μm), the highest Sq values (0.97 ± 0.15 μm), and the highest Sv values (4.37 ± 0.16 μm). The Pearson correlation test revealed positive correlations between the particle size (from control up to 110 μm) and Sa, Sq, or Sv parameters for all zirconia grades: Sa; *r* = 0.930 (*p* < 0.001) for 3Y-TZP, *r* = 0.928 (*p* < 0.001) for 4Y-PSZ, and *r* = 0.891 (*p* < 0.001) for 5Y-PSZ, Sq; *r* = 0.933 (*p* < 0.001) for 3Y-TZP, *r* = 0.930 (*p* < 0.001) for 4Y-PSZ, and *r* = 0.895 (*p* < 0.001) for 5Y-PSZ, or Sv; *r* = 0.939 (*p* < 0.001) for 3Y-TZP, *r* = 0.949 (*p* < 0.001) for 4Y-PSZ, and *r* = 0.924 (*p* < 0.001) for 5Y-PSZ.

SEM images showed changes in the surface morphology after sandblasting using different sizes of Al_2_O_3_ (Figure 4). Grain boundaries of the zirconia surface were observed for the sandblasted 3Y-TZP using 25 μm alumina sand, while the disappearance of grain boundaries was observed for all other sandblasted specimens. Surface damages, such as micro-cracks, plastic deformations, surface melting, and Al_2_O_3_ particle impingement following the abrasive process were detected. 

EDX analysis demonstrated that the yttrium (Y) content increased as the Y_2_O_3_ doping level in zirconia increased. After the Al_2_O_3_ air-abrasion, the EDX analysis revealed that the presence of aluminum (Al) was evident on the zirconia surfaces (0.93–2.16%): The highest concentrations of Al for each zirconia grade were 2.01, 2.13, and 2.16% for 3Y-90, 4Y-110, and 5Y-110, respectively.

### 3.3. Quantitative Determination of Zirconia Phases and Residual Stresses

Representative μRaman spectra are shown in Figure 6. The shapes of the spectra for the 3Y-TZP specimens were clearly different from those for 4Y- or 5Y-PSZ specimens. Figure 7 depicts the peak shift of the tetragonal band at around 147 cm^−1^ due to the stress induced at the zirconia surface after sandblasting as a function of Al_2_O_3_ particle size. For 3Y-TZP, the tetragonal peaks at 147, 456, and 641 cm^−1^ decreased, while the monoclinic peaks at 178 and 506 cm^−1^ increased after sandblasting and the amount of changes increased as the particle size increased (Figure 6A). As shown in Figure 7, the tetragonal peak at 147 cm^−1^ shifted to a higher wavenumber after sandblasting (*p* < 0.05). The peak shift increased with increasing particle size up to 125 μm (*p* < 0.05). The hump at the left shoulder of the tetragonal peak for the sandblasted specimen was observed and the size of this peak increased with increasing particle size.

For 4Y-PSZ and 5Y-PSZ, it was difficult to distinguish between the tetragonal and cubic phase at 147 cm^−1^ due to their overlapping wavenumbers. As the particle size increased, the tetragonal peak at 641 cm^−1^ decreased, whereas the cubic peak at 625 cm^−1^ was maintained (Figure 6B,C). The monoclinic peak at 178 cm^−1^ slightly increased with an increase in the particle size. As presented in Figure 7, the tetragonal peak at 147 cm^−1^ shifted to a higher wavenumber up to 90 μm sand for 4Y-PSZ (*p* < 0.05), while the peak shifted to a higher wavenumber up to 25 μm sand for 5Y-PSZ (*p* < 0.05). 

## 4. Discussion

This study investigated the effect of Al_2_O_3_ sandblasting particle size on the surface topography and residual compressive stresses of three different zirconia grades. The results of this study indicated that sandblasting with different Al_2_O_3_ sizes under a given pressure, time, distance, and impact angle (0.2 MPa, 10 s/cm^2^, 10 mm, and 90°, respectively) altered the surface topographies and induced compressive stresses. However, each zirconia grade showed a different dependence on the particle sizes in terms of surface topography, phase transformations, and compressive stresses. Therefore, the null hypotheses was rejected.

In this study, sandblasting using 25 μm Al_2_O_3_ sand did not significantly modify the surface topographies in most subgroups (*p* > 0.05). With a further increase in the particle size up to 110 μm, the scalar values of the surface texture parameters (Sa, Sq, and Sv) increased (*p* < 0.05). The substantial increase in the scalar values suggests that larger particles could offer better micro-interlocking with resin cements. In Zhao et al.’s study, a larger particle promoted the surface roughness of conventional as well as highly translucent zirconia, resulting in an increase in the shear bond strength [10]. Their study was conducted with two different particle sizes (50 and 110 μm). Unlikely, the result of this study revealed that the scalar values for sandblasted zirconia using 125 μm sand fell below those using 90 μm sand for all three zirconia grades (Figure 5). This may be attributed to the surface flattening, broadening, and material loss induced by the larger particles [38]. Furthermore, we detected loosely packed 125 μm particles in the SEM image (Figure 3E), which can lead to the reduced surface area due to decreasing agglomeration. Their wide particle size distribution (29.91–592.39 μm) may also contribute to the lower scalar values. The result of this study showed that the larger the particle size specification, the greater the dispersion. In terms of particle packing density, broadening the particle size distribution improves packing efficiency by allowing smaller particles to pack the spaces between the larger ones since larger particles pack less efficiently than smaller ones, creating bigger voids [39]. The result of the surface topography reported by Inokoshi et al. [33] was different from ours. They found that Al_2_O_3_ sandblasting did not significantly change the surface topographies of three different zirconia grades, but they only used 50 μm Al_2_O_3_ sand. Moreover, our results revealed that highly translucent zirconia groups (4Y- and 5Y-PSZ) were abraded faster than conventional 3Y-TZP, which were in agreement with the result of Zhao et al.’s study [10]. The highly translucent zirconia had weaker mechanical properties compared to conventional 3Y-TZP [6,7] and thus, less energy was required for the striking particles to alter the surface topography. 

The microscopic observation of the surfaces in this study showed the disappearance of grain boundaries after sandblasting (Figure 4), which could be attributed to the stress-related diffusion along grain boundaries [40]. The majority of abrasive particles exhibited sharp and grit shapes instead of spherical shapes (Figure 3), which could induce more stresses inside the particles and thus the particle fracture. The impingement of the particle fragments in the surface during the abrasive process (Figure 4L,R) could be responsible for the debonding of resin cements from the zirconia surface [12]. A larger particle gained higher kinetic energy with higher temperature during the impact process, resulting in a local surface melting on the zirconia surface (Figure 4K,P,Q), which could lead to the substantial damage of lower-strength highly translucent zirconia, although the micro-cracks resulting from sandblasting could exert a positive effect on the adhesion of resin cements [12]. Furthermore, alumina sand with sharp edges, rather than spherical shapes, can yield a reduced abraded area during the impact process. Thus, higher blasting pressure should be employed to obtain an abraded surface and the higher pressure can give rise to surface damages as well as deleterious effects on the adhesion of luting cements [12]. In this sense, the manufacturing techniques to produce abrasive particles with spherical shapes can be a crucial factor in the air abrasion process.

In this study, the phase transformation and related residual stresses after sandblasting were evaluated using μRaman spectroscopic techniques. In particular, the peak shift of the tetragonal band at around 147 cm^−1^ was traced to determine the residual stresses, where a tetragonal and monoclinic band were not overlapping [41]. For 3Y-TZP, sandblasting induced tetragonal to monoclinic phase transformation at the surface, and as a consequence, the related stresses upshifted the tetragonal band (*p* < 0.05), which was referred to as a blue shift (Figure 6 and Figure 7) [42]. The positive shift after sandblasting was consistent with those reported by the previous studies [17,40,43]. Im et al. explained that the Raman shift increased if the material lattice was compressed and gained energy [44]. In this study, a larger alumina particle exhibited a larger blue shift (*p* < 0.05), with a decreasing tetragonal peak and increasing monoclinic peak as the particle size increased. Therefore, a larger blue shift could indicate the presence of larger compressive stresses, resulting in the improved mechanical properties. In addition, a hump at the left shoulder of the tetragonal peak was detected with larger particles, suggesting that the induced residual stresses would be large enough to create lattice distortions [43] or the formation of a rhombohedral phase [33], contributing to the enhanced strength of zirconia. 

For 4Y-PSZ, a broad cubic peak was detected at 625 cm^−1^ [45]. The intensity of the cubic peak was relatively unchanged, while the tetragonal peak at 456 cm^−1^ decreased with increasing particle size. This must be related to the low transformability of the cubic phase [6]. Broadened tetragonal peaks with larger particles were also detected at 147 and 456 cm^−1^, assuming that this peak broadening resulted from the lattice distortion of tetragonal zirconia [43] or the presence of rhombohedral phase [33] as a consequence of sandblasting. As shown in Figure 7, the tetragonal peak at 147 cm^−1^ exhibited a blue shift and the amount of shift increased up to 90 μm (*p* < 0.05), whereas no further peak shifting was observed (*p* > 0.05). This can be explained by the increased critical stress concentration associated with the crack growth behavior in a residual stress field [40]. The formation of the compressive stresses may improve the strength of sandblasted 4Y-PSZ using up to 90 μm sand, hindering the propagation of the flaws. However, deleterious effects on the strength could be expected with larger alumina particles. 

For 5Y-PSZ, a larger particle created a bigger change in the surface topography, and the changes grew at a faster rate than the other zirconia grades (Figure 5). This finding can be attributed to the lowest flexural strength of 5Y-PSZ [7]. The μRaman spectra in this study indicated that there were no clear changes in the peak intensity after sandblasting (Figure 6C) due to the little potential for the phase transformation of cubic grain in 5Y-PSZ [6]. The tetragonal peak at 147 cm^−1^ was blue-shifted with 25 μm sand (*p* < 0.05), indicating the presence of residual compressive stresses. However, the μRaman peak exhibited a shift toward a shorter wavelength after sandblasting using 50 μm sand or larger (*p* < 0.05), which was referred to as a red shift [42] (Figure 7). As opposed to a blue shift, the downshift of the Raman peak position occurred if the material was under tensile strain [44]. A red shift would be related to the structural disorder associated with lattice defects [46] or a decrease in the phonon energies due to a lattice stretching [47]. In addition, increased temperature could cause a red shift due to increased thermal expansion in the crystal lattice [47]. Thus, the plastic deformation and surface melting on the surface of 5Y-PSZ observed in this study (Figure 4) would result from the internal tensile stress and increased temperature due to the impact load during the sandblasting. As a consequence, structural degradation would be expected after sandblasting for 5Y-PSZ. 

Within the range of sandblasting parameters investigated in this study, Al_2_O_3_ sandblasting altered the surface topographies of three different zirconia grades, but each grade showed different reactions to the changes in the particle sizes, which can be attributed to its unique physical, chemical, and mechanical properties. In consideration of the potential benefits from the residual stresses associated with phase transformation and any risks of mechanical failure, the use of 110 μm Al_2_O_3_ sand is recommended for a given pressure, time, distance, and impact angle (0.2 MPa, 10 s/cm^2^, 10 mm, and 90°, respectively) for 3Y-TZP. For 4Y-TZP, the use of 90 μm Al_2_O_3_ sand can be recommended in order to obtain a bigger change in the surface topography for a good bonding with minimal surface defects. For 5Y-TZP, sandblasting using 25 μm Al_2_O_3_ sand may be clinically preferred in order to prevent possible damage to the zirconia surface. However, there are some limitations to these recommendations. In this study, we focused on the changes in the Al_2_O_3_ particle sizes, while other parameters were kept constant. The effect of changes in other parameters on the surface topography and crystallography of three different zirconia grades should be added for future studies. In addition, the measurements of the flexural strength and bond strength of resin cement to zirconia for the tested groups were not performed here. Future research should investigate the influence of different sandblasting protocols on the mechanical properties or bonding performance of three different zirconia grades. In this study, we applied 3-D texture parameters (Sa, Sq, and Sv) which were height parameters according to the ISO 25178 standard to evaluate the topography of abraded zirconia surfaces. Further studies on the characterization of the surface topography should include a complementary set of additional parameters, such as spatial, volumetric, or feature parameters. The use of multi-scale analysis can allow a more accurate evaluation of the surface topography. 

## 5. Conclusions

This study presented a methodology for assessing the state of abraded zirconia surfaces using 3-D texture parameters (Sa, Sq, and Sv), but a complementary set of new parameters can be beneficial for the accurate evaluation of surface topographies. Following sandblasting, a Raman peak shift was detected and residual stresses associated with phase transformations were calculated for three dental zirconia grades. Therefore, this analytic technique can also be used to quantify the residual stresses on the highly translucent zirconia surfaces. Based on the experimental and analytical results on the surface topography and residual compressive stresses of three different dental zirconia grades after sandblasting, the following specific conclusion was drawn:Three different dental zirconia grades showed different degrees of dependence on the Al_2_O_3_ particle size in terms of surface topography, phase transformation, and internal compressive stresses.Al_2_O_3_ sandblasting modified the surface topographies of three different zirconia grades, although highly translucent zirconia showed more pronounced topographic changes compared to conventional zirconia.A larger particle induced more compressive residual stresses related to the t → m phase transformation for 3Y-TZP, while only 25 μm sand induced residual stresses due to little potential for the phase transformation of the cubic grain in 5Y-PSZ.Therefore, the results of this study suggested that the recommended Al_2_O_3_ particle sizes under given pressure, time, distance, and impact angle (0.2 MPa, 10 s/cm^2^, 10 mm, and 90°, respectively) were 110 μm sand for 3Y-TZP, 90 μm sand for 4Y-PSZ, and 25 μm sand for 5Y-PSZ in order to obtain a bigger change in the surface topography for a good bonding with minimal surface damage.

## Figures and Tables

**Figure 1 materials-14-00610-f001:**
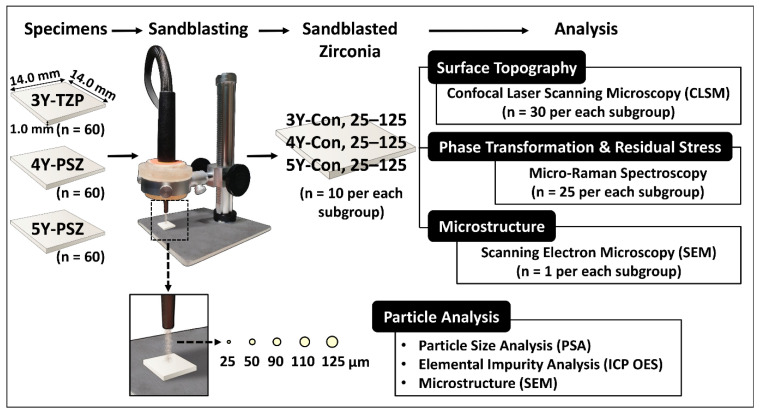
Flow chart of the experimental procedure; specimen preparation and Al_2_O_3_ sandblasting conditions as well as the analytical methods associated with surface topography, residual stresses, microstructure, and Al_2_O_3_ particle analysis are depicted. 3Y-TZP: 3 mol% yttria-stabilized tetragonal zirconia polycrystal; 4Y-PSZ: 4 mol% partially-stabilized zirconia; 5Y-PSZ: 5 mol% partially-stabilized zirconia; CLSM: confocal laser scanning microscopy; ICP OES: inductively coupled plasma optical emission spectrometry.

**Figure 2 materials-14-00610-f002:**
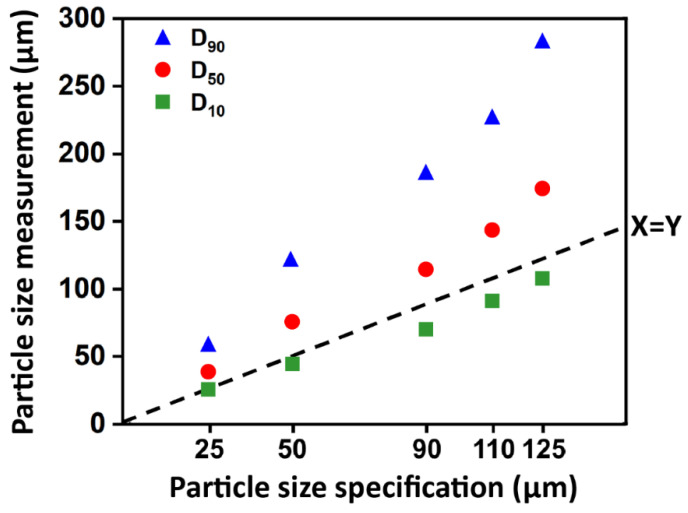
Characteristics of the alumina particle size distributions. There were significant differences among all particle size specifications and the measured particle sizes increased as the particle size specification increased (*p* < 0.05).

**Figure 3 materials-14-00610-f003:**
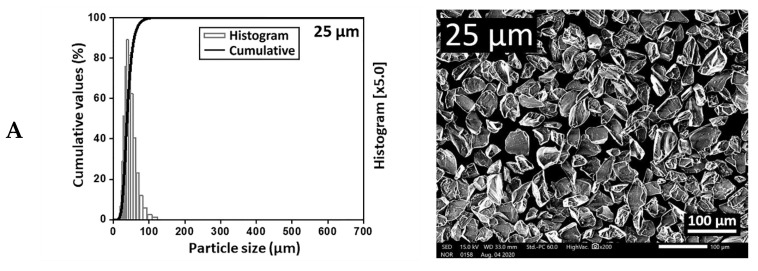
Particle size distributions and scanning electron micrographs of Al_2_O_3_ particles. The finest particles were observed in the 25 μm specification, while the largest particles prevailed in the 125 μm specification. Most of the particles had a grit shape with sharp edges and rarely a spherical shape. (**A**) 25 μm specification; (**B**) 50 μm specification; (**C**) 90 μm specification; (**D**) 110 μm specification; (**E**) 125 μm specification.

**Figure 4 materials-14-00610-f004:**
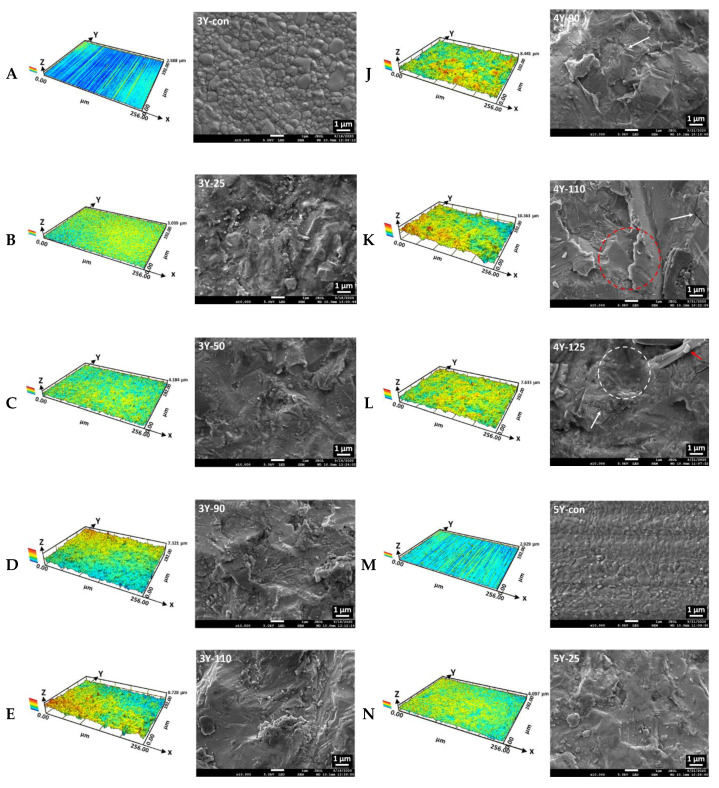
Three-dimensional representations obtained by confocal laser scanning microscopy and scanning electron microscopic images (magnification 10,000×) of each subgroup. For the SEM images, the white arrow indicates a micro-crack; the red arrow indicates Al_2_O_3_ particle debris deposited on the abraded zirconia surface; the white circle indicates plastic deformation; the red circle indicates surface melting. (**A**) 3Y-con.; (**B**) 3Y-25.; (**C**) 3Y-50.; (**D**) 3Y-90.; (**E**) 3Y-110.; (**F**) 3Y-125.; (**G**) 4Y-con.; (**H**) 4Y-25.; (**I**) 4Y-50.; (**J**) 4Y-90.; (**K**) 4Y-110.; (**L**) 4Y-125.; (**M**) 5Y-con.; (**N**) 5Y-25.; (**O**) 5Y-50.; (**P**) 5Y-90.; (**Q**) 5Y-110.; (**R**) 5Y-125.

**Figure 5 materials-14-00610-f005:**
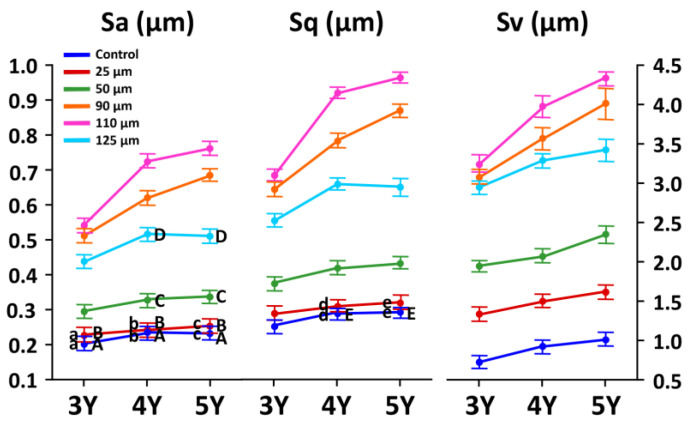
The surface texture parameters of each subgroup for three different zirconia grades. Mean values represented with same uppercase letters (within each row) or lowercase letters (within each column) are not significantly different based on the results of pairwire comparisons for simple main effects using Sidak adjustment (*p* > 0.05). The values of the Sa, Sq, or Sv parameters increased with an increase in the particle size up to 110 μm, whereas those values decreased with 125 μm alumina sand, lying between those with 50- and those with 90 μm alumina sand for all zirconia grades.

**Figure 6 materials-14-00610-f006:**
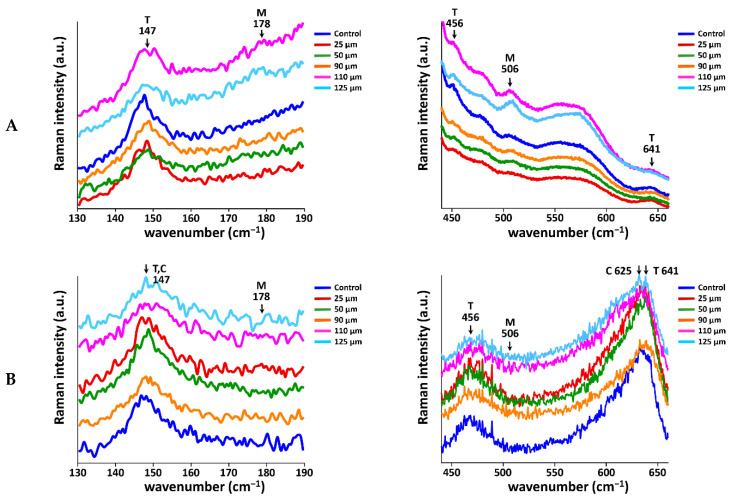
Representative μRaman spectra for each subgroup of three zirconia grades. (**A**) 3Y-TZP.; (**B**) 4Y-PSZ.; (**C**) 5Y-PSZ. The shapes of the spectra for 3Y-TZP specimens are clearly different from those for 4Y- or 5Y-PSZ specimens. In order to confirm the crystalline structures of the specimens, Raman peaks from 130 to 190 cm^−1^ and from 440 to 660 cm^−1^ are shown. For 3Y-TZP, the tetragonal peaks at 147, 456, and 641 cm^−1^ decreased, while the monoclinic peaks at 178 and 506 cm^−1^ increased as the particle size increased. For 4Y- and 5Y-PSZ, the tetragonal peaks at 641 cm^−1^ decreased, whereas the cubic peaks at 625 cm^−1^ were maintained.

**Figure 7 materials-14-00610-f007:**
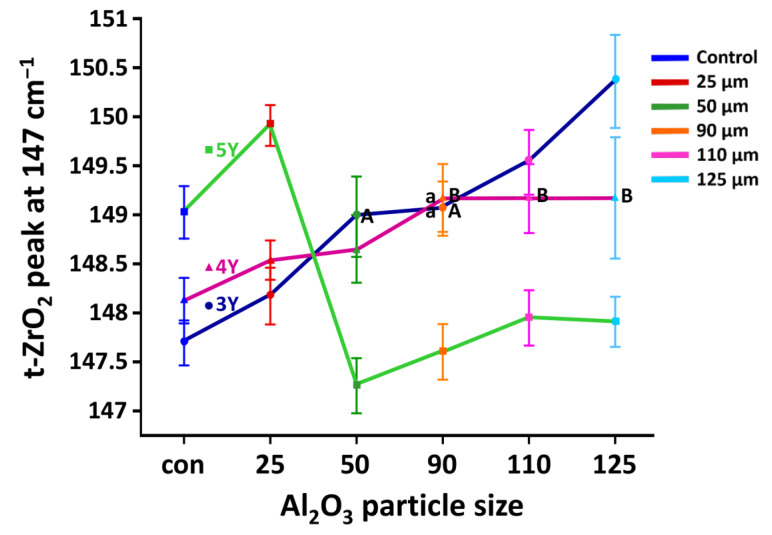
The 147 cm^−1^ of the t-ZrO_2_ Raman peak shift for each subgroup of three zirconia grades as a function of Al_2_O_3_ particle size. Mean values represented with the same uppercase letters (within each zirconia grade) or lowercase letters (within each particle size) are not significantly different based on the results of pairwire comparisons for simple main effects using the Sidak adjustment (*p* > 0.05). For 3Y-TZP, the peak shift increased with increasing particle sizes up to 125 μm (*p* < 0.05). The tetragonal peak at 147 cm^−1^ shifted to a higher wavenumber up to 90 μm for 4Y-PSZ (*p* < 0.05), while the peak shifted to a higher wavenumber up to 25 μm for 5Y-PSZ (*p* < 0.05).

**Table 1 materials-14-00610-t001:** Characteristics of the zirconia systems investigated.

Materials	Manufacturer	Shade	Batch No.	Sintering	Composition ^a^	Flexural Strength (MPa) ^a^	Toughness (MP am^1/2^) ^a^
KATANA ML	Kuraray Noritake	A Light	EASLS	1500 °C for 2 h	3Y-TZP (<15% c) ^b^	900–1100	3.5–4.5
KATANA STML	Kuraray Noritake	A2	EAVHC	1550 °C for 2 h	4Y-PSZ (>25% c) ^b^	600–800	2.5–3.5
KATANA UTML	Kuraray Noritake	A2	DZVML	1550 °C for 2 h	5Y-PSZ (>50% c) ^b^	500–600	2.2–2.7

^a^ Values reported by Zhang and Lawn [37]; ^b^ cubic (c) phases in zirconia.

**Table 2 materials-14-00610-t002:** Concentrations of trace elements in Al_2_O_3_ sandblasting particles obtained by ICP OES.

Element	25 μm	50 μm	90 μm	110 μm	125 μm
**Si**					
Mean (mg/kg)	259.971	322.217	356.979	173.206	545.963
SD (mg/kg)	0.39	1.87	0.61	.076	3.71
RSD (%)	0.15	0.58	0.17	0.44	0.68
LOD	0.018				
**Fe**					
Mean (mg/kg)	40.695	63.193	45.943	38.530	80.348
SD (mg/kg)	0.22	0.47	0.15	0.19	0.78
RSD (%)	0.54	0.75	0.33	0.49	0.97
LOD	0.002				
**Zr**					
Mean (mg/kg)	1.657	3.270	3.684	1.842	5.645
SD (mg/kg)	0.03	0.04	0.03	0.03	0.01
RSD (%)	1.82	1.21	0.54	1.53	0.03
LOD	0.001				

SD: standard deviation of the mean; RSD: relative standard deviation; LOD: limit of determination.

## Data Availability

The data presented in this study are available on request from the corresponding author.

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
