# Peer review of "Effect of Al2O3 Sandblasting Particle Size on the Surface Topography and Residual Compressive Stresses of Three Different Dental Zirconia Grades"

_materials, 2021, doi:10.3390/ma14030610_

Round 1

Reviewer 1 Report

The current paper studies the sandblasting process and the effect of the sand particle size on the surface finish and stresses when sandblasting several types of zirconia . The roughness metric measured and studied here is a 3D which was measured using 3D optical microscopy. The author use SEM to determine the residual stresses, the particles used in sandblasting are made from Al2O3 and five different sizes were used. Authors employ ANOVA to find significance of particles size on the studied outputs.

Line 70 stacking many references to state simple facts is not recommended.

The literature review is well written, and English is clear. However, it is recommended to expand more on the literature review and discuss more past studies on sandblasting of zirconia or similar materials using Al2O3 particles. Discuss what they have done and what were their main findings and how your current work differs from theirs.

Please discuss in short about figure 1 although we can see all details clearly but it is recommended to add some details too.

What is the reason for the choice of those specific particles sizes? Where they just randomly selected or based on past studies or recommended industry practice? Please justify

There is some issue in lines 109-112 (text overlapping table)!

Line 122 what is special about this specimen holder?

Please explain more about line 176-177 and the test techniques used in the study, explain what it is and why it is used (what is its importance)?

Figure 2 is interesting however, there is no clear explanation of why this trend is observed, why the 125 microns have the greatest dispersion, please justify and explain this phenomena and for other particle sizes too.

Line 197 how does the particle shape affect the sandblasting process, please explain and use references where appropriate

Figure 4 is not acceptable in its current form, it spans for more than three pages!

Please consider keeping one or two and creating graphs with bar charts to better show us the results observed from all those images and elemental analysis.

SEM images are too small in Figure 4 and is difficult to come to any conclusions from looking at them with their current size.

It appears that the surface roughness Sa is non linear with particle size, as we can see particles with 90-110 microns showed highest roughness that 125 microns, the authors need to explain this using scientific evidence from past literature, perhaps explain the mechanism of sandblasting in the text and refer how the particle size and shape might affect surface roughness accordingly.

In Figure 5 the change in Sa is from 0.2 to 0.75, does this range of change is considered significant? For example, would a surface finish of 0.2 would be much different for certain applications of zirconia than that with Sa of 0.8, please explain and justify.

now I see that discussion was added later in the paper, perhaps it is better to combine the discussion with previous section so that the authors discuss and comment on each figure at the same time rather than having to go backward and forward to check the explanations on observed results.

Author Response

We, the authors, highly appreciate the detailed valuable comments on this manuscript.

The suggestions are quite helpful for us and we incorporate them in the revised paper.

The revision was listed below the comments and recommendations one by one.

================================================================

Response to Reviewer 1 Comments

The current paper studies the sandblasting process and the effect of the sand particle size on the surface finish and stresses when sandblasting several types of zirconia. The roughness metric measured and studied here is a 3D which was measured using 3D optical microscopy. The author use SEM to determine the residual stresses, the particles used in sandblasting are made from Al2O3 and five different sizes were used. Authors employ ANOVA to find significance of particles size on the studied outputs.

Line 70 stacking many references to state simple facts is not recommended.

- We changed the references in the revised manuscript.

The literature review is well written, and English is clear. However, it is recommended to expand more on the literature review and discuss more past studies on sandblasting of zirconia or similar materials using Al2O3 particles. Discuss what they have done and what were their main findings and how your current work differs from theirs.

- We expanded more on the literature review and discussed more past studies in the revised manuscript.

Please discuss in short about figure 1 although we can see all details clearly but it is recommended to add some details too.

- We added some details in Figure 1 in the revised manuscript: Figure 1. Flow chart of the experimental procedure; specimen preparation and Al2O3 sandblasting conditions as well as the analytical methods associated with surface topography, residual stresses, microstructure, and Al2O3 particle analysis are depicted.

What is the reason for the choice of those specific particles sizes? Where they just randomly selected or based on past studies or recommended industry practice? Please justify

- We selected commercially available Al2O3 particles on the dental market (25, 50, 90, 110, and 125 μm) and we explained it in the revised manuscript.

There is some issue in lines 109-112 (text overlapping table)!

- We corrected them in the revised manuscript. Thank you.

Line 122 what is special about this specimen holder?

- It was a custom-made specimen holder designed by the authors (Figure 1). We explained it in the revised manuscript.

Please explain more about line 176-177 and the test techniques used in the study, explain what it is and why it is used (what is its importance)?

- We described more about the test techniques in the revised manuscript: A two-way ANOVA was performed to determine the effect of two independent variables (yttria content; 3 mol% yttria for 3Y-TZP, 4 mol% yttria for 4Y-PSZ, and 5 mol% yttria for 5Y-PSZ and alumina particle size) (1) on the surface topography and (2) on the Raman wavenumber of the tetragonal zirconia (t-ZrO2) band at around 147 cm-1. The interactions between two independent variables were verified and pairwise comparisons for simple main effects of independent variables were analyzed by using SPSS syntax. In addition, the Pearson correlations between particle size and surface characterization parameters of the subgroups for all zirconia grades were analyzed.

Figure 2 is interesting however, there is no clear explanation of why this trend is observed, why the 125 microns have the greatest dispersion, please justify and explain this phenomena and for other particle sizes too.

- We explained it in the Discussion section: The result of this study showed that the larger the particle size specification, the greater the dispersion. In terms of particle packing density, broadening the particle size distribution improves packing efficiency by allowing smaller particles to pack the spaces between larger ones since larger particles pack less efficiently than smaller ones, creating bigger voids [35].

Line 197 how does the particle shape affect the sandblasting process, please explain and use references where appropriate

- We explained it in the Discussion section: Furthermore, alumina sand with sharp edges, rather than spherical shapes, can yield reduced abraded area during the impact process. Thus, higher blasting pressure should be employed to get an abraded surface and the higher pressure can give rise to surface damages as well as deleterious effects on the adhesion of luting cements [12]. In this sense, the manufacturing techniques to produce abrasive particles with spherical shapes can be a crucial factor in the air abrasion process.

Figure 4 is not acceptable in its current form, it spans for more than three pages!

- We changed Figure 4 in the revised manuscript.

Please consider keeping one or two and creating graphs with bar charts to better show us the results observed from all those images and elemental analysis.

- Thank you for your advice.

Reviewer 2 Report

General remarks:

Please apply style and formatting which are coherent with MDPI template (MS Word).

Abstract:

“For each subgroup, surface roughness was evaluated using 3D confocal laser microscopy and surface topography was analyzed with scanning electron microscope.”

I would rather say that, in general, surface topography is measured by 3D confocal laser microscopy. The term roughness is scale-dependent. It is most often determined by filtration of the original measurement. There are some standardized approaches which can provide a hint of what wavelength should be considered as cut-off. However, the division between roughness and waviness is usually subjective to particular characteristic of a measured surface. There is a good publication that deal with that problem:

Brown, Christopher A., et al. "Multiscale analyses and characterizations of surface topographies." CIRP annals 67.2 (2018): 839-862, doi: 10.1016/j.cirp.2018.06.001.

SEM usually allows only a visual inspection of the images. “Surface texture” and “surface topography” are the terms usually considered as having the same meaning. Please do bear that in mind when revising this paper as a whole.

I would suggest rewriting this sentence in the following manner:

“For each subgroup, surface topography was evaluated using 3D confocal laser microscopy and inspected with scanning electron microscope.”

In the abstract I would expect some quantifiable findings related to surface topography. Now it is “Al2O3 sandblasting improved the surface roughness of…”. The notion of “better” might be differently understood by readers.

Introduction:

There is very little about the originality and importance of this study. The motivation also could be better presented.

The paper covers an analysis of surface topography but it is not well-covered in the introduction. You seem to use and advanced instrumentation to capture fine scale surface features but use an analysis which tells very little about this complexity. There are numerous parameters which can better describe the topographic effects of the discussed formation process. These are covered in ISO 25178. I would focus on studies which cover feature parameters. In addition, it seems that feature-based characterization might also be a good try for your study (see: https://doi.org/10.1016/j.addma.2020.101273 or similar). Other approach is multiscale analysis like here: https://doi.org/10.1016/j.cirp.2018.06.001. Finally there is a motif analysis which can help to determine the measures of geometric features like holes and valleys which were created by sandblasting. See e.g. https://doi.org/10.1016/S0890-6955(99)00118-2 and some more examples of this method used in similar studies.

Please try to include these examples (and more) in your literature review. Please try to expand methods and result sections with more complex analysis. For particle analysis and many other methods as above there is a free trial license of MountainsMap (Digital Surf) software available which can help you in this study. This is a state-of-the-art tool used for the analysis of surface topographies.

Materials and Methods

Please change the terms in Figure 1 according to my previous comments (surface roughness).

This is unclear: “Only the polished surface received the sandblasting treatment”. Is the polished side of each specimen?

There are different materials presented in Table 1. Could you please expand on their differences? What is their structure, chemical/physical structure and so on…?

Usually in sand-blasting there is  atrace amount of Al2O3 evident when visually inspect the as-blasted samples under the microscope. These are microparticles which are stuck in the material. There is nothing about it here. I do not see them on microscopic images. Were there not present at all? Please explain.

Some references to already conducted studies related to the Abrasive Blasting Particle analysis should be provided.

As pointed before: “The 3D surface roughness parameter,Sa (arithmetic mean deviation) was measured using a 147software (LEXT-OLS, version 6.0.3, Olympus, Tokyo, Japan).” Sa is just a basic parameter which describe the average deviation of the heights to the mean plane. There is a multitude of better suited parameters and methods which might be used here.

There is also nothing here about how the measurements were processed. If form was removed and how? Were outliers removed and how? Was the noise filtered? What was the cut-off wavelength? These are basic information that is missing in this study. The authors insist on the term roughness but do not decompose waviness from roughness.

“…Sidak adjustment was conducted to examine the effect of yttria…” There is nothing on yttria mentioned before.

Results:

The first sentence “A normal distribution and homogeneity of variances were verified with Shapiro-Wilk test and Levene 182test, respectively (p< 0.05).” should in the previous section

There is a lot about the particle analysis but the resulted surface roughness is not well analyzed with respect to the geometry of those particles. This is an important issue which could reveal the nature of formation process-surface texture interactions.

Some statistical correlations (if possible) should be given between certain surface characterization parameters (not only Sa) and particle sizes for all studied materials.

Discussion and conclusion:

I would expect an improvement here when more complex surface texture analysis and statistical correlations should be given.

Author Response

We, the authors, highly appreciate the detailed valuable comments on this manuscript.

The suggestions are quite helpful for us and we incorporate them in the revised paper.

The revision was listed below the comments and recommendations one by one.

================================================================

Response to Reviewer 2 Comments

General remarks:

Please apply style and formatting which are coherent with MDPI template (MS Word).

- Yes, we applied style and formatting which are coherent with MDPI template.

Abstract:

“For each subgroup, surface roughness was evaluated using 3D confocal laser microscopy and surface topography was analyzed with scanning electron microscope.”

I would rather say that, in general, surface topography is measured by 3D confocal laser microscopy. The term roughness is scale-dependent. It is most often determined by filtration of the original measurement. There are some standardized approaches which can provide a hint of what wavelength should be considered as cut-off. However, the division between roughness and waviness is usually subjective to particular characteristic of a measured surface. There is a good publication that deal with that problem:

Brown, Christopher A., et al. "Multiscale analyses and characterizations of surface topographies." CIRP annals 67.2 (2018): 839-862, doi: 10.1016/j.cirp.2018.06.001.

SEM usually allows only a visual inspection of the images. “Surface texture” and “surface topography” are the terms usually considered as having the same meaning. Please do bear that in mind when revising this paper as a whole.

I would suggest rewriting this sentence in the following manner:

“For each subgroup, surface topography was evaluated using 3D confocal laser microscopy and inspected with scanning electron microscope.”

- We changed the terms and rewrote the sentences in the revised manuscript as a whole.

In the abstract I would expect some quantifiable findings related to surface topography. Now it is “Al2O3 sandblasting improved the surface roughness of…”. The notion of “better” might be differently understood by readers.

- We added some quantifiable findings related to surface topography in the revised manuscript: 5Y-PSZ abraded with 110-μm sand showed the highest Sa value (0.76 ± 0.12 μm).

- We rewrote it in the revised manuscript: Al2O3 sandblasting altered the surface topographies of three zirconia grades (p < 0.05),~

Introduction

There is very little about the originality and importance of this study. The motivation also could be better presented.

The paper covers an analysis of surface topography but it is not well-covered in the introduction. You seem to use and advanced instrumentation to capture fine scale surface features but use an analysis which tells very little about this complexity. There are numerous parameters which can better describe the topographic effects of the discussed formation process. These are covered in ISO 25178. I would focus on studies which cover feature parameters. In addition, it seems that feature-based characterization might also be a good try for your study (see: https://doi.org/10.1016/j.addma.2020.101273 or similar). Other approach is multiscale analysis like here: https://doi.org/10.1016/j.cirp.2018.06.001. Finally there is a motif analysis which can help to determine the measures of geometric features like holes and valleys which were created by sandblasting. See e.g. https://doi.org/10.1016/S0890-6955(99)00118-2 and some more examples of this method used in similar studies.

Please try to include these examples (and more) in your literature review. Please try to expand methods and result sections with more complex analysis. For particle analysis and many other methods as above there is a free trial license of MountainsMap (Digital Surf) software available which can help you in this study. This is a state-of-the-art tool used for the analysis of surface topographies.

- We included more studies on the surface topographies in the revised manuscript: Surface topography can influence the geometrical, physical, and chemical properties of the material [19]. In the analysis of surface texture, areal based three-dimensional analyses, defined within ISO 25178, can provide more precise information than two-dimensional profile measurements [20]. A number of studies dealt with the characterization methods for surface topographies. The feature-based characterization, a recently introduced method, could offer spatial information in relation to orientation of topographical properties of the surfaces [20]. Another systematic approach is multiscale analyses, which include the characterization of surface topographies at multiple scales of observation to control the different interactions of the scales [19]. A new 3D-motif method can be applied to determine the geometric features like holes and valleys which are created by sandblasting [21]. Reflectance confocal microscopy based on the focus detection method is one of the optical metrology techniques. Noncontact optical measurement methods have been widely used for their great reliability and flexibility, but the noise of the light source can affect the measurement quality [22]. A novel spiral-scanning laser differential confocal measurement method can help to reduce the existing disturbance [23].

Materials and Methods

Please change the terms in Figure 1 according to my previous comments (surface roughness).

- We changed the term in Figure 1 in the revised manuscript.

This is unclear: “Only the polished surface received the sandblasting treatment”. Is the polished side of each specimen?

- We changed ‘surface’ to ‘side’ in the revised manuscript.

There are different materials presented in Table 1. Could you please expand on their differences? What is their structure, chemical/physical structure and so on…?

- We expanded Table 1 in the revised manuscript (composition, flexural strength, toughness).

Usually in sand-blasting there is a trace amount of Al2O3 evident when visually inspect the as-blasted samples under the microscope. These are microparticles which are stuck in the material. There is nothing about it here. I do not see them on microscopic images. Were there not present at all? Please explain.

- We described those microparticles stuck in the zirconia surface in the revised manuscript: the red arrow indicates Al2O3 particle debris deposited on the abraded zirconia surface (Figure 4L and 4R)

Some references to already conducted studies related to the Abrasive Blasting Particle analysis should be provided.

- We provided the reference in the revised manuscript:

  • Hallmann, L.; Ulmer, P.; Reusser, E.; Hämmerle, C.H.F. Effect of blasting pressure, abrasive particle size and grade on phase transformation and morphological change of dental zirconia surface. Surf. Coat. Technol. 2012, 206, 4293-4302. https://doi.org/10.1016/j.surfcoat.2012.04.043

As pointed before: “The 3D surface roughness parameter,Sa (arithmetic mean deviation) was measured using a 147software (LEXT-OLS, version 6.0.3, Olympus, Tokyo, Japan).” Sa is just a basic parameter which describe the average deviation of the heights to the mean plane. There is a multitude of better suited parameters and methods which might be used here.

- We provided the scalar values of texture parameters in the revised manuscript: Sa and Sq

There is also nothing here about how the measurements were processed. If form was removed and how? Were outliers removed and how? Was the noise filtered? What was the cut-off wavelength? These are basic information that is missing in this study. The authors insist on the term roughness but do not decompose waviness from roughness.

- We described how the measurements were processed in the revised manuscript: Surface measurements were processed with the form and outlier eliminated. Tilt was corrected and a 3-D surface was constructed with the distance to the optical center in the X-axis, the tilt angle on the Y-axis, and the flatness error on the Z-axis. A robust short wavelength pass Gaussian filter (cut-off wavelength: 10 μm) was applied to the data in order to decompose waviness from roughness.

“…Sidak adjustment was conducted to examine the effect of yttria…” There is nothing on yttria mentioned before.

- We rewrote it in the revised manuscript: A two-way ANOVA was performed to determine the effect of two independent variables (yttria content; 3 mol% yttria for 3Y-TZP, 4 mol% yttria for 4Y-PSZ, and 5 mol% yttria for 5Y-PSZ and alumina particle size) ~

Results:

The first sentence “A normal distribution and homogeneity of variances were verified with Shapiro-Wilk test and Levene 182test, respectively (p< 0.05).” should in the previous section

- We moved the sentence to the Materials and Methods section in the revised manuscript.

There is a lot about the particle analysis but the resulted surface roughness is not well analyzed with respect to the geometry of those particles. This is an important issue which could reveal the nature of formation process-surface texture interactions.

- We described it in the Discussion section in the revised manuscript: Furthermore, alumina sand with sharp edges, rather than spherical shapes, can yield reduced abraded area during the impact process. Thus, higher blasting pressure should be employed to get an abraded surface and the higher pressure can give rise to surface damages as well as deleterious effects on the adhesion of luting cements [12]. In this sense, the manufacturing techniques to produce abrasive particles with spherical shapes can be a crucial factor in the air abrasion process.

Some statistical correlations (if possible) should be given between certain surface characterization parameters (not only Sa) and particle sizes for all studied materials.

- We described it in the Result section in the revised manuscript: The Pearson correlation test revealed positive correlations between particle size (from control up to 110 μm) and Sa or Sq parameters for all zirconia grades: Sa; r = 0.930 (p < 0.001) for 3Y-TZP, r = 0.928 (p < 0.001) for 4Y-PSZ, and r = 0.891 (p < 0.001) for 5Y-PSZ, Sq; r = 0.933 (p < 0.001) for 3Y-TZP, r = 0.930 (p < 0.001) for 4Y-PSZ, and r = 0.895 (p < 0.001) for 5Y-PSZ.

Discussion and conclusion:

I would expect an improvement here when more complex surface texture analysis and statistical correlations should be given.

- We tried to provide more complex surface texture analysis and statistical correlations in the revised manuscript.

Round 2

Reviewer 1 Report

authors have answered all the comments and paper can be accepted

Author Response

We, the authors, highly appreciate the detailed valuable comments on this manuscript.  Thank you.

Sincerely,

Hee-Kyung Kim, DDS MSD PhD

Reviewer 2 Report

Thank you very much for improving this paper and considering my comments.

My final concern is about the following issue:

Some statistical correlations (if possible) should be given between certain surface characterization parameters (not only Sa) and particle sizes for all studied materials.

We described it in the Result section in the revised manuscript: The Pearson correlation test revealed positive correlations between particle size (from control up to 110 μm) and Sa or Sq parameters for all zirconia grades: Sa; r = 0.930 (p < 0.001) for 3Y-TZP, r = 0.928 (p < 0.001) for 4Y-PSZ, and r = 0.891 (p < 0.001) for 5Y-PSZ, Sq; r = 0.933 (p < 0.001) for 3Y-TZP, r = 0.930 (p < 0.001) for 4Y-PSZ, and r = 0.895 (p < 0.001) for 5Y-PSZ.

Using Sq is a little step forward to more complex analysis. Since you analyze surface topographies which appear to look as a series of overlapping craters I would suggest to incorporate in your analysis also Sv parameter (maximum pit height according to ISO 25178). Also volume parameters are suitable here: Vvc, core void volume and Vvv, valley void volume. It would be great to see additional plots like Fig. 5 for the aforementioned parameters.

Author Response

We, the authors, highly appreciate the detailed valuable comments on this manuscript.

The suggestions are quite helpful for us and we incorporate them in the revised paper.

The revision was listed below the comments and recommendations one by one.

================================================================

Response to Reviewer 2 Comments

My final concern is about the following issue:

Some statistical correlations (if possible) should be given between certain surface characterization parameters (not only Sa) and particle sizes for all studied materials.

We described it in the Result section in the revised manuscript: The Pearson correlation test revealed positive correlations between particle size (from control up to 110 μm) and Sa or Sq parameters for all zirconia grades: Sa; r = 0.930 (p < 0.001) for 3Y-TZP, r = 0.928 (p < 0.001) for 4Y-PSZ, and r = 0.891 (p < 0.001) for 5Y-PSZ, Sq; r = 0.933 (p < 0.001) for 3Y-TZP, r = 0.930 (p < 0.001) for 4Y-PSZ, and r = 0.895 (p < 0.001) for 5Y-PSZ.

Using Sq is a little step forward to more complex analysis. Since you analyze surface topographies which appear to look as a series of overlapping craters I would suggest to incorporate in your analysis also Sv parameter (maximum pit height according to ISO 25178). Also volume parameters are suitable here: Vvc, core void volume and Vvv, valley void volume. It would be great to see additional plots like Fig. 5 for the aforementioned parameters.

-------------------------------------------------------------------------------------------------------------

We incorporated Sv parameter and changed Figure 5 in the revised manuscript. Unfortunately, we did not measured the functional volume parameters in this study. We feel sorry that we cannot provide either Vvc or Vvv.

Material & Methods

The areal texture parameters were obtained using a software (LEXT-OLS, version 6.0.3, Olympus, Tokyo, Japan): Sa, the arithmetic mean height; Sq, the root mean square height; and Sv, the maximum pit height of the scale-limited surface according to ISO 25178 [20].

Result

5Y-PSZ abraded with 110-μm sand showed the highest Sa values (0.76 ± 0.12 μm), the highest Sq values (0.97 ± 0.15 μm), and the highest Sv values (4.37 ± 0.16 μm). The Pearson correlation test revealed positive correlations between particle size (from control up to 110 μm) and Sa, Sq, or Sv parameters for all zirconia grades: Sa; r = 0.930 (p < 0.001) for 3Y-TZP, r = 0.928 (p < 0.001) for 4Y-PSZ, and r = 0.891 (p < 0.001) for 5Y-PSZ, Sq; r = 0.933 (p < 0.001) for 3Y-TZP, r = 0.930 (p < 0.001) for 4Y-PSZ, and r = 0.895 (p < 0.001) for 5Y-PSZ, or Sv; r = 0.939 (p < 0.001) for 3Y-TZP, r = 0.949 (p < 0.001) for 4Y-PSZ, and r = 0.924 (p < 0.001) for 5Y-PSZ.

Figure 5. The surface texture parameters of each subgroup for three different zirconia grades. Mean values represented with same uppercase letters (within each row) or lowercase letters (within each column) are not significantly different based on the results of pairwire comparisons for simple main effects using Sidak adjustment (p > 0.05). The values of Sa, Sq, or Sv parameters increased with an increase in the particle size up to 110 μm, whereas those values decreased with 125-μm alumina sand, lying between those with 50- and those with 90-μm alumina sand for all zirconia grades.
